# Prevalence of HTN and its risk factors among tribal population in Chhattisgarh (India) as per the fifth round of the National Family Health Survey

**Tripti Nagaria**[1☯], **Neha Singh**[2☯], **Madhur Verma**[3], **Angad Singh**[4], **Kamlesh Jain**[5], **NirmalVerma**[5], **Monika Dengani**[5], **Shailendra Agrawal**[5], **Sukhbir Singh**[6], **Sonu Goel**[7]*

**1** Department of Obstetrics and Gynaecology, Pt. JNM Medical College, Raipur, Chhattisgarh, India, **2** Virology Lab, Department of Microbiology, Pt. JNM Medical College, Raipur, Chhattisgarh, India, **3** Department of Community and Family Medicine, All India Institute of Medical Sciences, Bathinda, Punjab, India, **4** Department of Biostatistics and Epidemiology, International Institute for Population Science, Mumbai, India, **5** Department of Community Medicine, Pt. JNM Medical College, Raipur, Chhattisgarh, India, **6** Poulation Research Center, Punjab University, Chandigarh, India, **7** Department of Community Medicine and School of Public Health, Post Graduate Institute of Medical Education and Research, Chandigarh, India

☯ Authors contributed equally to this work and are both first authors.
* sonu.goel@ul.ie

## Abstract

### Background

The so-called protected tribal population also faces the burden of non-communicable diseases (NCDs). The high-altitude tribes are thought to be genetically and environmentally protected from hypertension-like diseases.

### Methods

Survey of the National Family Health Survey-5 (2019-2022) employs a two-stage sampling method (PSU). The total population of 72010, aged 15 years and above, were included from Chhattisgarh, India, out of which 27760 belong to tribes. Pre-hypertension (pre-HTN) and hypertension (HTN) were measured using standard procedures.

### Results

Overall prevalence of pre-HTN and HTN (42.7% and 26%) in tribes was comparable to the general population (43.14% and 25.48%). The prevalence of pre-HTN was higher than HTN in tribes. We observed a positive association between increasing age and prevalence of HTN in males and females. aOR ratio (12.58; 9.72 -16.28) among the females aged 65-74 years was the highest, and the aOR (9.63; 6.89 -13.44) ratio among the males was recorded highest at 75 years and above. Odds of developing HTN increased with higher education (aOR 1.24; 1 -1.53 and 0.69; 0.5 -0.94), highest wealth index (aOR 1.38; 1.04 -1.83 and 1.35; 1.01 -1.8) in male and females respectively.

**Data availability statement:** All relevant data are within the manuscript and its Supporting Information files.

**Funding:** The author(s) received no specific funding for this work.

**Competing interests:** The authors have declared that no competing interests exist.

## Conclusion

The elevated frequencies of HTN among tribes against their background condition confirm that their epidemiological transition is loaded with significant HTN. We recommend sharing this initiative with our primary healthcare providers and to the policymakers to take corrective measures because they serve as the first point of contact for early screening to reduce the burden of illness.

## Introduction

Hypertension (HTN) is one of the significant risk factors for cardiovascular diseases (CVDs) and is established as a global health burden [1]. In India, HTN is the most prevalent non-communicable disease (NCD) and a significant public health issue [2]. As a part of its Sustainable Development Goals (SDG), the Government of India is committed to a one-third reduction in premature mortality due to NCDs by 2030, more than half of which is contributed by CVDs [3,4]. It is estimated that up to a third of CVD deaths can be avoided by proper control of HTN [5]. India has also committed to reducing the prevalence of HTN to 25% of the 2010 level by 2025 in its National NCD Monitoring Framework [6]. As such, the implementations are primarily directed towards the urban areas with more affluent strata of society and tribal populations are neglected due to the notion that HTN is the problem of the wealthy and affluent classes in developing countries. In India, scheduled tribes (ST) constitute 8.6% of the population, with an overwhelmingly diverse range of types [7,8]. The ST communities are identified as culturally or ethnographically unique by the Indian constitution and marginalized with relatively poorer health indicators [9]. Existing literature focuses on HTN in urban and rural areas of India, but rare reports are available concerning the tribal population of Chhattisgarh. Currently, HTN is an emerging public health problem in the tribal people. There needs to be more scientific knowledge about the existence of HTN in the ST population of Chhattisgarh. It is believed that tribal culture carries the heritage of traditional healing methods to address mind and body, and due to their minimalist lifestyle, it is thought that they are immune to HTN [10,11]. Faith in the natural healing system strongly influences health practices, health-seeking behavior, and the choices of tribal people. HTN research conducted worldwide has concluded that tribal populations have a lower prevalence and their BP does not increase with aging [12–14]. This claim provides a fascinating epidemiological window for the study of HTN. In the context of Chhattisgarh, tribal populations are confined to isolated highland terrains and deep forests and are linked to poverty, illiteracy, and starvation [15]. In a survey on the tribe population of Chhattisgarh, the majority of hypertensive were unaware or undiagnosed for the disease [16]. Standard measurements to ascertain the rule of halves among tribal populations depicted poor health-seeking behavior in the ST population due to low levels of awareness and knowledge and an inaccessible health system [16]. Hypertensive in tribal people can be attributed to persistent undernutrition during childhood, worsened by unhealthy lifestyles in later stages of their lives [17]. A study depicted a 37% prevalence of HTN among the adult tribal population of Kerala and its significant association with age, gender, education, wealth index, physical inactivity, alcohol consumption, and overweight/obesity [18]. Tribal populations are not frequently involved in scientific studies because of their scattered habitats, inaccessible terrain, and nomadic living nature [19]. With the research-gap as mentioned above, we believe that there is a compelling need to assess the prevalence of HTN among the tribal community of Chhattisgarh and the factors associated by utilizing the data retrieved from the fifth round of the National Family Health Survey (NFHS-5). The survey is designed to offer estimates on various critical indicators concerning HTN at

the national and Sub-national levels. The present study is crucial to address the gaps in our current understanding regarding the epidemiological transitions among the vulnerable populations of India, including the tribal population. It will sensitize the policymakers and public health administrators in this regard.

## Methodology

Study design: We did a secondary analysis of the National Family Health Survey (NFHS)-5 (2019-2021) data [8].

Study setting: Chhattisgarh, a state of India, has 33 administrative districts. It is a heavily forested state in central India known for its temples and waterfalls. It is bounded by the Indian states of Uttar Pradesh and Jharkhand to the north and northeast, Odisha to the east, Telangana to the south, and Maharashtra and Madhya Pradesh to the west. Raipur is the Capital of Chhattisgarh, a state with an area of 135,192 square km and a population of 2.94 crores (Fig 1)

Study population: The survey included a tribal population of Chhattisgarh aged 15 years and above. The NFHS-5 collects information based on background characteristics such as age, urban, rural, education, marital status, religious wealth index, smoking inside the house, smokers or tobacco users, and alcohol consumption. Blood pressure (BP) was measured for all women and men aged 15 and above using an Omron Blood Pressure Monitor to determine the prevalence of HTN.

Sample size and sampling technique: The NFHS-5 survey employs a two-stage sampling method (PSUs). To choose villages and Census Enumeration Blocks as the major sampling units in rural and urban areas, the probability proportional to size (PPS) sampling technique was employed in the survey to select communities and CEBs in each rural and urban stratum. The data collection approach, including a description of a sample of houses, questionnaire delivery, and validation, is fully documented elsewhere [7,14]. From NFHS-5, a total population from person files of 2843917. For this study, a sample, 2078315, aged 15 years and above, was taken because information related to HTN was not taken for the below 15 years. Out of 2078315, the total sample in the Chhattisgarh state population was recorded as 79745, but blood pressure measurements were done only for 72010, out of which 27760 were tribal (Fig 2).

### Data variables and data sources

Outcome variables: HTN was our primary dependent variable. It was measured using an Omron Blood Pressure Monitor three times with an interval of five minutes between readings. HTN was defined as a systolic BP level of 140 mm Hg or higher or a diastolic BP level of 90 mm Hg or higher, as per JNC-7 [20]. Antihypertensive medication was taken to meet the criteria of HTN in the current study. A systolic BP range of 121-90 mmHg or a diastolic BP range of 81-89 mmHg were considered pre-hypertension.

Independent variables: Independent variables for this analysis was set with different age and range such as (15-25, 25-34, 35-44, 45-54, 51-60, 65-74, 75 and above), residence (urban, rural), education (no education/pre-school, primary, secondary, higher), marital status (never married, recently married, widow/divorced/separated), religion (Hindu and others), wealth index (poorest, poorer, middle, richer, richest), smoke inside the house (never, daily, weekly, in a months or less), smokers or tobacco users (Y/N), alcohol consumption (Y/N).

Data analysis: Analysis was done using STATA 15 (Stata Corp, College Station, TX, USA. We used descriptive statistics and bivariate statistics for analysis. Categorical variables were reported in percentage and compared using Chi-square tests. Continuous variables were presented as mean ± standard deviation (SD) for normally distributed data. A multivariate logistic regression model was used in our study to adjust for potential and established risk

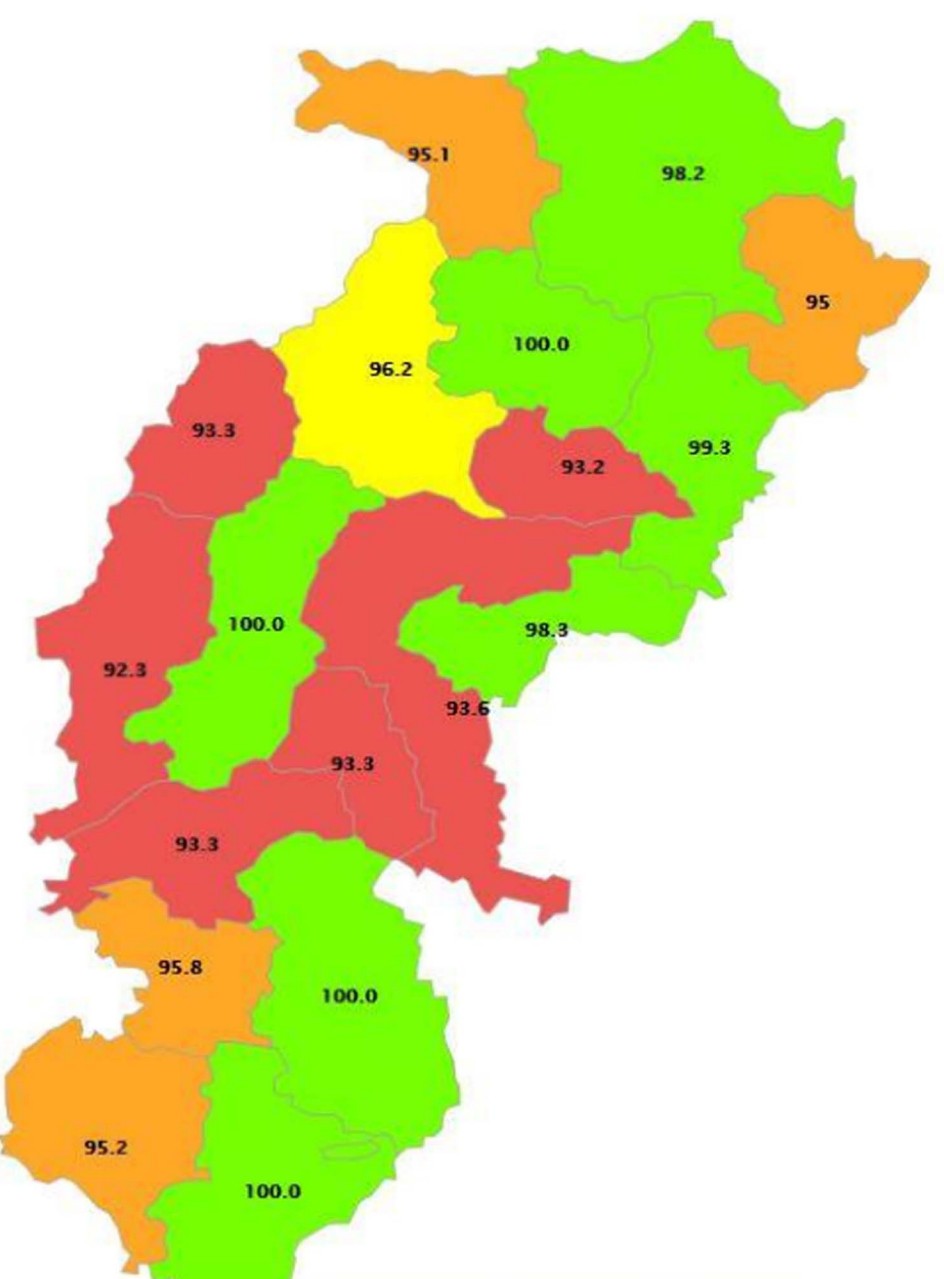

**Fig 1. Map of Chhattisgarh, India.** This is the <u>Fig 1</u> Chhattisgarh is a tribal heartland of India, preserving rich cultural heritage and traditions.

<u>https://doi.org/10.1371/journal.pone.0318268.g001</u>

factors for HTN in the tribal population. Adjusted Odds Ratio (aOR) for the risk of HTN was recorded. All statistical tests were two-tailed, and the cut-off of the significant level was defined as P < 0.05.

Ethical proclamation: This is a secondary analysis of the publicly accessible, freely available NFHS-5 (2019-2021) dataset collected from a nationally representative survey. Standard protocols were used to get the information from the Demographic Health Survey website <u>http://rchiips.org/nfhs/NFHS-5Reports/NFHS-5_INDIA_REPORT.pdf</u>. The ethical approval

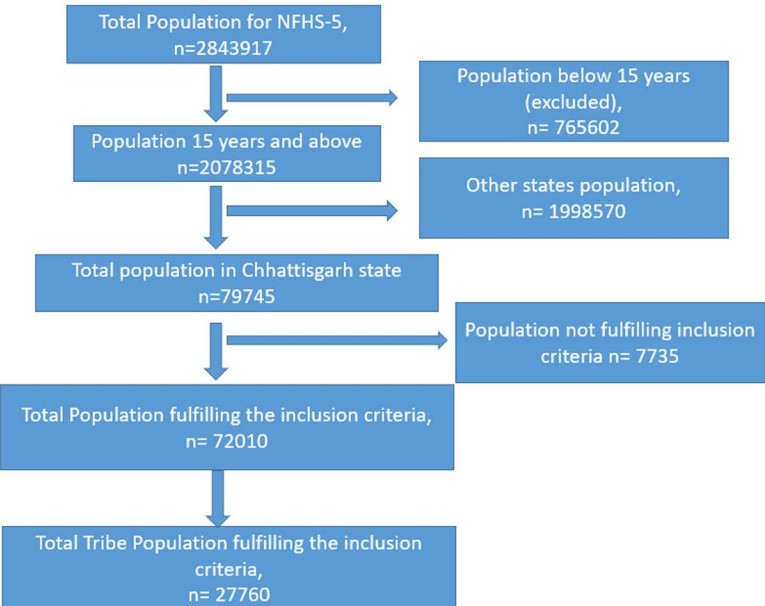

**Fig 2. Sample size selection.** This is the Fig 2 Flow chart depicting the sample selection process for tribe population of Chhattisgarh from NFHS-5 survey, India.

for the NFHS-5 surveys is obtained from the ethics review board of the International Institute for Population Sciences (IIPS), Mumbai, India and approval for secondary data was obtained from The Institute's Ethical Committee of the Postgraduate Institute of Medical Education and Research (PGIMER), Chandigarh ethically approved the study (IEC-08/2022-2535 dated 17.08.2022). Participants consent not required as secondary data was used for this publication.

## Results

Blood pressure measurements were done for the general population of 72010 from the state of Chhattisgarh. Of these 72010, 27760 were from scheduled tribes and were included in our analysis, aged 15 and above who had given consent for blood pressure readings [see Figs 1, 2]. Table 1 depicts the distribution of the participants based on their background characteristics such as age, urban, rural, education, marital status, religion wealth index, and smoke inside the house, smokers or tobacco users, alcohol consumption. Of the total participants included in the analysis, 47.0% (13031) were male.

Prevalence of pre-HTN (pre-HTN) among study participants is compared with the overall prevalence of pre-HTN in Chhattisgarh and is exhibited in Table 2. Pre-HTN was among the ST (42.7%) was comparable to the prevalence in the general population (43.14%). Pre-HTN was slightly higher (46.95%) in males than females (38.93%). In males prevalence of pre-HTN was recorded highest (52.68%) in 25-34 years, with higher education (50.63%) and belonging to highest wealth quintile (52%). Lowest pre-HTN was observed in the oldest ST males over 75 years. In ST females, prevalence of pre-HTN was highest (46.97% and 45.2%) between 35-44 years and, minimum years of education (i.e., upto primary class) respectively. Lowest pre-HTN was recorded in the oldest females (between 65-74 years).

Prevalence of HTN (HTN) among ST population of Chhattisgarh and general population is illustrated in Table 3. Prevalence of HTN in ST population was estimated as 26% compared to 25.48% in the general population, it was 28.22% and 24.03% respectively among the ST

**Table 1. Sample characteristics of the tribal population aged 15 and above in Chhattisgarh, India, NFHS-5, 2019-21.**

| Background characteristics | Male | Female | Total |
|---|---|---|---|
| Total | 47.0 (13031) | 53.0 (14729) | 100% (27760) |
| **Age** | % (N) | % (N) | % (N) |
| 15-25 | 25.6 (7204) | 25 (3286) | 26.1 (3918) |
| 25-34 | 21.6 (6250) | 21.7 (2931) | 21.5 (3319) |
| 35-44 | 17.3 (4862) | 16.8 (2289) | 17.7 (2573) |
| 45-54 | 15.3 (4315) | 15.5 (2039) | 15.1 (2276) |
| 51-60 | 12 (3155) | 12.5 (1528) | 11.6 (1627) |
| 65-74 | 6 (1469) | 6.4 (721) | 5.7 (748) |
| 75 and above | 2.2 (505) | 2.2 (237) | 2.3 (268) |
| **Residence** | | | |
| Urban | 8.6 (2283) | 8.3 (1035) | 8.9 (1248) |
| Rural | 91.4 (25477) | 91.7 (11996) | 91.2 (13481) |
| **Education** | | | |
| No education/Pre-school | 34.6 (11016) | 22.9 (3650) | 44.9 (7366) |
| Primary | 15.6 (4136) | 18.7 (2393) | 12.8 (1743) |
| Secondary | 43.5 (11108) | 50.7 (6126) | 37.1 (4982) |
| Higher | 6.3 (1490) | 7.6 (857) | 5.2 (633) |
| **Marital status** | | | |
| Never married | 24.2 (6800) | 27.5 (3589) | 21.3 (3211) |
| Currently married | 66.1 (18294) | 67.8 (8824) | 64.6 (9470) |
| Widowed/Divorced/Separated | 9.7 (2660) | 4.7 (616) | 14.1 (2044) |
| **Religion** | | | |
| Hindu | 95.8 (26836) | 95.7 (12596) | 95.8 (14240) |
| Others | 4.2 (924) | 4.3 (435) | 4.2 (489) |
| **Wealth index** | | | |
| Poorest | 52.1 (16470) | 52.2 (7756) | 52.1 (8714) |
| Poorer | 25.5 (6068) | 25.8 (2854) | 25.2 (3214) |
| Middle | 12.1 (2840) | 11.9 (1319) | 12.2 (1521) |
| Richer | 6.8 (1670) | 6.6 (770) | 6.9 (900) |
| Richest | 3.6 (712) | 3.5 (332) | 3.6 (380) |
| **Smoke inside the house** | | | |
| Never | 35.9 (9840) | 34.8 (4471) | 36.9 (5369) |
| Daily | 25 (6531) | 25.7 (3155) | 24.3 (3376) |
| Weekly | 17.1 (4797) | 17 (2278) | 17.1 (2519) |
| In a months or less | 22.1 (6592) | 22.4 (3127) | 21.8 (3465) |
| **Smoke or uses of tobacco** | | | |
| No | 59.5 (15678) | 45.6 (5700) | 71.7 (9978) |
| Yes | 40.6 (12070) | 54.4 (7319) | 28.3 (4751) |
| **Drinks Alcohol** | | | |
| No | 71.3 (18532) | 52.6 (6450) | 87.8 (12082) |
| Yes | 28.7 (9215) | 47.4 (6569) | 12.2 (2646) |

male and females. Highest prevalence (59.61% and 48.49%) was seen in male tribes ≥ 65. The prevalence was higher (44.25%) in male tribes who were living alone (widow, divorced or separated). Similarly in ST females, HTN was higher in older age groups.

**Table 2. Pre-HTN tribal population of Chhattisgarh and its comparison with total population NFHS-5, India, 2019-2021.**

| Background characteristics | Scheduled Tribes | | | General population total % |
|---|---|---|---|---|
| | Male % | Female % | Total % | |
| **Total** | **46.95 (13031)** | **38.93 (14729)** | **42.7 (27760)** | **43.14 (72010)** |
| **Age** | | | | |
| 15-25 | 47.24 | 32.4 | 39.22 | 38.73 |
| 25-34 | 52.68 | 41.49 | 46.77 | 47.63 |
| 35-44 | 50.4 | 46.97 | 48.54 | 49.48 |
| 45-54 | 45.03 | 42.95 | 43.94 | 44.45 |
| 51-60 | 40.94 | 36.08 | 38.45 | 40.03 |
| 65-74 | 40.52 | 31.47 | 35.96 | 35.61 |
| 75 and above | 26.56 | 34.15 | 30.73 | 29.37 |
| **Residence** | | | | |
| Urban | 46.42 | 36.9 | 41.23 | 43.25 |
| Rural | 47 | 39.13 | 42.84 | 43.1 |
| **Education** | | | | |
| No education/Pre-school | 42.55 | 40.06 | 40.84 | 41.02 |
| Primary | 47.59 | 45.2 | 46.55 | 45.32 |
| Secondary | 48.18 | 36.09 | 42.71 | 43.07 |
| Higher | 50.63 | 34.29 | 43.53 | 45.3 |
| **Marital status** | | | | |
| Never married | 48.12 | 32.74 | 40.95 | 40.91 |
| Currently married | 46.7 | 41.49 | 44 | 44.6 |
| Widowed/Divorced/Separated | 43.78 | 36.54 | 38.19 | 38.1 |
| **Religion** | | | | |
| Hindu | 47.08 | 38.98 | 42.78 | 43.18 |
| Others | 44.23 | 37.81 | 40.87 | 41.84 |
| **Wealth index** | | | | |
| Poorest | 46.38 | 39.86 | 42.93 | 42.64 |
| Poorer | 47.85 | 38.74 | 43.07 | 43.03 |
| Middle | 48.67 | 36.23 | 42 | 43.39 |
| Richer | 42.22 | 38.03 | 39.95 | 43.68 |
| Richest | 52 | 37.78 | 44.38 | 43.34 |
| **Smoke inside the house** | | | | |
| Never | 48.1 | 38 | 42.6 | 42.66 |
| Daily | 46.28 | 40.07 | 43.08 | 42.37 |
| Weekly | 45.85 | 39.39 | 42.42 | 44.79 |
| In a months or less | 46.78 | 38.91 | 42.67 | 43.83 |
| **Smoke or uses of tobacco** | | | | |
| No | 47.14 | 38.33 | 41.5 | 42.28 |
| Yes | 46.78 | 40.47 | 44.45 | 45.1 |
| **Drinks Alcohol** | | | | |
| No | 47.45 | 38.89 | 41.86 | 42.09 |
| Yes | 46.3 | 39.26 | 44.71 | 47.43 |

**Table 3. HTN tribal population of Chhattisgarh and its comparison with total population NFHS-5, India, 2019-2021.**

| Background characteristics | Scheduled Tribes (Weighted %) | | | % General population total |
|---|---|---|---|---|
| | Male | Female | Total | |
| **Total** | **28.22 (13031)** | **24.03 (14729)** | **26.00 (27760)** | **25.48 (72010)** |
| **Age** | | | | |
| 15-25 | 10.01 | 5.75 | 7.7 | 7.01 |
| 25-34 | 20.91 | 12.55 | 16.5 | 15.22 |
| 35-44 | 30.63 | 24.36 | 27.23 | 25.87 |
| 45-54 | 39.06 | 36.92 | 37.94 | 37.62 |
| 51-60 | 44.98 | 48.22 | 46.64 | 46.84 |
| 65-74 | 48.49 | 54.34 | 51.43 | 51.49 |
| 75 and above | 59.61 | 53.39 | 56.19 | 56.88 |
| **Residence** | | | | |
| Urban | 30.97 | 24.64 | 27.51 | 25.37 |
| Rural | 27.97 | 23.97 | 25.85 | 25.51 |
| **Education** | | | | |
| No education/Pre-school | 38.5 | 36.72 | 37.27 | 38.05 |
| Primary | 32.67 | 22.53 | 28.25 | 30.66 |
| Secondary | 22.21 | 10.92 | 17.1 | 17.97 |
| Higher | 26.08 | 11.92 | 19.93 | 20.53 |
| **Marital status** | | | | |
| Never married | 11.26 | 7.52 | 9.51 | 8.91 |
| Currently married | 34 | 24.96 | 29.32 | 28.96 |
| Widowed/Divorced/Separated | 44.25 | 44.75 | 44.64 | 44.75 |
| **Religion** | | | | |
| Hindu | 27.94 | 23.82 | 25.76 | 25.35 |
| Others | 34.4 | 28.83 | 31.48 | 29.18 |
| **Wealth index** | | | | |
| Poorest | 28.68 | 25.09 | 26.78 | 26.77 |
| Poorer | 26.3 | 22.27 | 24.19 | 23.72 |
| Middle | 25.11 | 23.72 | 24.36 | 24 |
| Richer | 35.15 | 21.12 | 27.57 | 25.3 |
| Richest | 32.87 | 27.59 | 30.04 | 28.48 |
| **Smoke inside the house** | | | | |
| Never | 28.15 | 24.78 | 26.31 | 25.87 |
| Daily | 27.32 | 23.66 | 25.43 | 24.98 |
| Weekly | 30.2 | 23.85 | 26.83 | 25.74 |
| In a months or less | 27.87 | 23.3 | 25.48 | 24.82 |
| **Smoke or uses of tobacco** | | | | |
| No | 23.37 | 20.4 | 21.47 | 22.15 |
| Yes | 32.3 | 33.23 | 32.65 | 33.32 |
| **Drinks Alcohol** | | | | |
| No | 23 | 22.29 | 22.54 | 23.61 |
| Yes | 34.09 | 36.5 | 34.64 | 33.36 |

**Table 4. Adjusted Odds Ratio (aOR) of HTN among male, female tribes population of Chhattisgarh aged 15 and above, by their background characteristics, NFHS-5, India, 2019-2021.**

| Background characteristics | Scheduled Tribes Male | Scheduled Tribes Female |
|---|---|---|
| | AOR (95% CI) | AOR (95% CI) |
| **Number of participants** | **13006** | **14719** |
| **Age** | | |
| 15-25 | | |
| 25-34 | 1.69***(1.39 -2.06) | 1.94***(1.58 -2.39) |
| 35-44 | 2.81***(2.27 -3.49) | 3.72***(3.01 -4.61) |
| 45-54 | 4.26***(3.43 -5.29) | 6.79***(5.45 -8.46) |
| 51-60 | 6.18***(4.93 -7.75) | 10.22***(8.12 -12.85) |
| 65-74 | 6.32***(4.92 -8.13) | 12.58***(9.72 -16.28) |
| 75 and above | 9.63***(6.89 -13.44) | 11.78***(8.49 -16.33) |
| **Residence** | | |
| Urban | | |
| Rural | 0.89 (0.75 -1.05) | 0.83**(0.7 -0.99) |
| **Education** | | |
| No education/Pre-school | | |
| Primary | 0.95 (0.85 -1.07) | 0.96 (0.84 -1.11) |
| Secondary | 1.13**(1 -1.26) | 0.81***(0.71 -0.94) |
| Higher | 1.24**(1 -1.53) | 0.69**(0.5 -0.94) |
| **Marital status** | | |
| Never married | | |
| Currently married | 1.63***(1.36 -1.96) | 1.2*(0.98 -1.47) |
| Widowed/Divorced/Separated | 1.6***(1.24 -2.06) | 1.35**(1.07 -1.69) |
| **Religion** | | |
| Hindu | | |
| Others | 1.29**(1.03 -1.6) | 1.42***(1.14 -1.77) |
| **Wealth index** | | |
| Poorest | | |
| Poorer | 0.95 (0.86 -1.06) | 1.04 (0.93 -1.16) |
| Middle | 0.98 (0.84 -1.14) | 1.11 (0.96 -1.29) |
| Richer | 1.18*(0.97 -1.43) | 0.89 (0.73 -1.1) |
| Richest | 1.38**(1.04 -1.83) | 1.35**(1.01 -1.8) |
| **Smoke inside the house** | | |
| Never | | |
| Daily | 0.97 (0.86 -1.09) | 1.01 (0.9 -1.13) |
| Weekly | 1.04 (0.92 -1.18) | 1 (0.88 -1.13) |
| In a months or less | 0.94 (0.84 -1.06) | 0.96 (0.85 -1.07) |
| **Smoke or uses of tobacco** | | |
| No | | |
| Yes | 0.88**(0.79 -0.98) | 1.04 (0.94 -1.15) |
| **Drinks Alcohol** | | |
| No | | |
| Yes | 1.26***(1.14 -1.4) | 1.04 (0.93 -1.17) |

CI: Confidence Interval; aOR-Adjusted Odds Ratio, *less significant (p value <0.1), ** Significant (p value <0.05), ***highly significant (p value <0.01).

We depict the predictors of HTN among male, and female tribal population using binary logistics regression analysis as exhibited in Table 4. We observed an increase in the odds ratio with age in both male and female. Highest ratio was recorded as (aOR 9.63; 6.89 -13.44) in male with the age of 75 and above years and (aOR12.58; 9.72 -16.28) in female with the age of between 65-74 years. Significantly higher odds were reported in males with higher education (aOR 1.24; 95% 1 -1.53), male currently married and divorced/separate (aOR 1.63; 1.36 -1.96); (aOR1.6; 1.24 -2.06), richest (aOR 1.38; 1.04 -1.83), smokers or tobacco users (aOR 0.88; 0.79 -0.98) and alcohol users (aOR 1.26; 1.14 -1.4) were found to be linked to a significant level of HTN. Similarly, in female tribes currently married and divorced/ separate status (aOR 1.35; 1.07 -1.69); (aOR 1.2; 0.98 -1.47), rich class (aOR 1.35;1.01 -1.83) significantly predicted HTN.

## Discussion

There is evidence that HTN has had negative health implications not only in India but world-wide since the early 19th century [20,21]. The tribal population is thought to be genetically and environmentally protected from HTN-like diseases [22]. According to prior research, it has been established that the tribes demonstrate a lower prevalence of HTN and that their BP does not increase with age due to the direct relationship between nature and a clean atmosphere [23]. The tribes of Chhattisgarh are a distinct race that lives primarily in deep jungles. These indigenous people have a distinct way of life that is infused with traditional rituals and traditions [24]. To the best of our knowledge, there has never been a study to estimate the prevalence of HTN among Chhattisgarhi tribals using large-scale data. There is negligible focus on tribal HTN, according to research before the year 2000; the comparatively shoddier prevalence of HTN among tribal populations has fascinated researchers around the globe [25]. A study of the adult Kurichias tribe in Kerala revealed a very low prevalence of HTN in 1999 (2.7%) [26]. A study carried out among the aboriginal Nicobarese tribe in 2010 revealed a very high prevalence (50.5%) of HTN [27,28]. Another study by Kusuma et al. from the Visakhapatnam district of Andhra Pradesh in 2004, showed that the prevalence of HTN was low among a primitive tribal group and high among a tribal population that was assimilating, indicating that acculturation is a factor in elevated blood pressure and the prevalence of HTN [15]. Hence, even indigenous tribes in the modern world have been influenced by NCDs such as HTN which is due to their poor lifestyle. Studies on two tribal populations in South India found a wide range of HTN, with 21.7% hypertensive in one group and 40% hypertensive in the other [29]. Another study on tribals in Central India discovered a 23% prevalence of HTN [30]. Acculturation has been identified as one of the causes of elevated hypertension among tribals in studies [16,19]. Furthermore, excessive alcohol intake among males (78%) may indicate one of the prevalence factors of HTN in the population. As a result, assessing the health of these communities is critical. Due to such facts, the present assessment was conducted to estimate the prevalence of HTN status in the tribal population of Chhattisgarh, India, using data from NFHS-5. One of the present study's strengths is using a large sample size. Despite the fact that our study was done in a tribal population, the prevalence of pre-HTN and HTN was found to be similar to the prevalence reported in the general Chhattisgarh population. Several research conducted in India in recent decades has shown that HTN is on the rise in both urban and rural groups [2,17,19,29–31]. There are also misconceptions that unhealthy lifestyles are only for the wealthy and urban inhabitants, whereas tribal people, who live a more traditional lifestyle, do not suffer from lifestyle ailments such as HTN. However, our study found that HTN also plagues tribals, and our findings are consistent with those reported by Laxmaiah et al. from tribals in Madhya Pradesh and Chhattisgarh [26]. The prevalence of HTN among adult tribals is nearly identical to the frequency reported for rural adults in Madhya Pradesh by the Integrated Disease Surveillance Project study [32]. Similarly,

the prevalence of HTN was discovered to be high among tribal populations in Bihar, India. They appear to be equally affected by the changing lifestyles brought about by migration, acculturation, and urbanization, which overshadows their heavy work mindset [33]. Genetic variables are thought to play a substantial role in the disparities in the prevalence of cardio-vascular illness and death between populations. Their effects, however, are difficult to discern. We found that certain factors such as increasing age, higher education, marital status, alcohol consumption, and smoking were significantly associated with HTN in the tribal population which is the same as the general population reported somewhere [34]. It is generally known that HTN increases with age. In the United Kingdom, stroke and coronary heart disease are the leading causes of death in adults over the age of 65. HTN is the most prevalent treated risk factor and treatment for hypertension can considerably reduce the risk of stroke and its recur-rence [35,36]. Some people's blood pressure drops as they get older due to illnesses including Alzheimer's and other forms of dementia, cancer, or reduced ventricular function after myo-cardial infarction. It is well known that smoking and alcohol are also powerful cardiovascular risk factors [37,38], The chance of having an ischemic stroke increases significantly with the consumption of three or more drinks per day, and myocardial infarction risk may also rise with heavier drinking [19]. In current findings, men had a higher prevalence of HTN than did women. However, gender disparities in HTN condition revealed that females are more likely to be hypertensive than males after the age of 45-50 years. However, after the age of 60, women were more likely to have HTN than men [15]. Other factors that could contribute to HTN include a deprived lifestyle, poor diet, and nutrition, and lack of healthcare facilities [27][2]. Most of the tribe's population remains unaware that they have pre-HTN or HTN and do not get proper facilities and disease diagnoses on time. In India, it is reported to have a fine facility and infrastructure deficit regarding healthcare facilities, mostly in tribal areas and Union Territories [26]. Acculturation/modernization also might be one of the components causing this elevation of HTN, but there might be other unmeasured elements that need more investigation.

Policymakers should consider how tribal populations' shifting health requirements are changing. With 8.6% of the population being tribal, India has difficulty bridging the health-care gap between tribal and non-tribal residents. Nearly triple disease burden affects the tribal people, including communicable and non-communicable illnesses, hunger, mental illness, and addictions exacerbated by poor health-seeking behavior. An expert committee on tribal health has made suggestions in response to growing requirements to close the current gap in the health status of tribal people at the latest by the year 2027 [28]. To create new knowledge and gain a deeper understanding of a disease or a health condition, the National Institute of Research in Tribal Health (NIRTH-ICMR), the center for advanced research, encourages in-depth investigation of a selected research problem like NCDs. Activities could concentrate on a single feature or several of them, such as cause, advancement, management, and preven-tion in the tribe population [39]. The findings of the NFHS-5 survey showed that there is a significant prevalence of HTN in the adult tribal population of Chhattisgarh, India, which is comparable to the frequency seen in both urban and rural populations as well as in both sexes. Thus, we can argue that tribal people are also vulnerable to non-communicable diseases like HTN and that urgent initiatives for its management are needed for them.

There are certain recommendations emerging from our study. Stakeholders must immedi-ately implement therapeutic and long-term preventive strategies to address the clearly rising trend in the prevalence of HTN in tribal communities. Although acculturation appears to be the main cause of the increase, other underlying causes are at play. To more effectively explain the developing tendency, a thorough grasp of these components is required. Altitude, physical activity, social variables, economic considerations, food, genetic factors, behavioral

factors, body type, population age structure, and access to healthcare are among these and are significant.

To conclude, we observe reducing disparities in the prevalence of HTN between tribes and the general population of Chhattisgarh. Other than HTN, the fact is that tribes struggle with more recent health issues in addition to common ailments must be widely acknowledged [34]. The fact that tribes struggle with more recent health issues in addition to common diseases must be widely acknowledged. Through several programs and creating a specific ministry for tribal affairs, the Indian government has vowed to progress these underprivileged people. The study's conclusions will inform concerned policymakers about the evolving healthcare requirements of tribal groups in India, not only in Chhattisgarh.

## Author contributions

**Conceptualization:** Sonu Goel.

**Data curation:** Neha Singh, Angad Singh, Monika Dengani, Shailendra Agrawal, Sonu Goel.

**Formal analysis:** Angad Singh.

**Methodology:** Madhur Verma, Sonu Goel.

**Project administration:** Sonu Goel.

**Resources:** Kamlesh Jain, Nirmal Verma, Shailendra Agrawal, Sukhbir Singh, Sonu Goel.

**Validation:** Neha Singh, Madhur Verma, Angad Singh.

**Visualization:** Sonu Goel.

**Writing – original draft:** Tripti Nagaria, Neha Singh.

**Writing – review & editing:** Tripti Nagaria, Neha Singh, Madhur Verma, Kamlesh Jain, Nirmal Verma.

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
