## [Decision Letter · Decision Letter 0]

2 Jun 2024

PONE-D-23-38296Prevalence of HTN and its risk factors among tribal population in Chhattisgarh, India as per the fifth round of the National Family Health SurveyPLOS ONE

Dear Dr. Goel,

Thank you for submitting your manuscript to PLOS ONE. After careful consideration, we feel that it has merit but does not fully meet PLOS ONE’s publication criteria as it currently stands. Therefore, we invite you to submit a revised version of the manuscript that addresses the points raised during the review process.

We look forward to receiving your revised manuscript.

Kind regards,

Pijush Kanti Khan, Ph.D.

Academic Editor

PLOS ONE

- DOI: 10.4103/ijmr.IJMR_2097_20

In your revision ensure you cite all your sources (including your own works), and quote or rephrase any duplicated text outside the methods section. Further consideration is dependent on these concerns being addressed.

4. We note that Figure 2 in your submission contain [map/satellite] images which may be copyrighted. All PLOS content is published under the Creative Commons Attribution License (CC BY 4.0), which means that the manuscript, images, and Supporting Information files will be freely available online, and any third party is permitted to access, download, copy, distribute, and use these materials in any way, even commercially, with proper attribution. For these reasons, we cannot publish previously copyrighted maps or satellite images created using proprietary data, such as Google software (Google Maps, Street View, and Earth). For more information, see our copyright guidelines: http://journals.plos.org/plosone/s/licenses-and-copyright.

1. You may seek permission from the original copyright holder of Figure 2  to publish the content specifically under the CC BY 4.0 license.  

Reviewers' comments:

Reviewer's Responses to Questions

**Comments to the Author**

1. Is the manuscript technically sound, and do the data support the conclusions?

Reviewer #1: Yes

Reviewer #2: Yes

Reviewer #3: Partly

2. Has the statistical analysis been performed appropriately and rigorously? 

Reviewer #1: Yes

Reviewer #2: Yes

Reviewer #3: No

3. Have the authors made all data underlying the findings in their manuscript fully available?

Reviewer #1: Yes

Reviewer #2: Yes

Reviewer #3: Yes

4. Is the manuscript presented in an intelligible fashion and written in standard English?

Reviewer #1: Yes

Reviewer #2: No

Reviewer #3: Yes

5. Review Comments to the Author

Reviewer #1: This study examines the prevalence of hypertension (HTN) and its risk factors among the tribal population in Chhattisgarh state using data from NFHS 5. I recommend several recommendations to improve the study's clarity and significance:

1. A paragraph explaining why the focus is specifically on Chhattisgarh state. This will help contextualize the study's geographic scope.

2. Given that numerous studies have already utilized NFHS 5 data to examine hypertension and its risk factors, add a paragraph that outlines the unique contributions of this research.

3. In the data analysis section, it is reported that no chi-square, mean, or standard deviation (SD) values were found using the respective statistical tests for categorical and continuous data. This needs verification and correction if necessary.

4. The study's title and objectives are centered on the tribal population of Chhattisgarh state. However, Tables 1 and 4 focus on tribal groups, whereas Tables 2 and 3 make comparisons with the general population. It is unclear why these comparisons are made. Consider revising the study’s objectives and title to reflect a comparative analysis between tribal and non-tribal populations of Chhattisgarh state, or alternatively, focus solely on tribal data from NFHS 4.

5. While Table 2 addresses the prevalence of prehypertension and Table 3 deals with hypertension, Table 4 only discusses hypertension without presenting an adjusted odds ratio for prehypertension. Given the high prevalence of prehypertension noted in the results, it would be beneficial to include a multivariate analysis for prehypertension in both sexes separately.

These suggestions aim to enhance the study’s coherence and depth, ensuring it makes a clear, valuable contribution to the existing body of knowledge.

Reviewer #2: The study reports on the prevalence of hypertension and its risk factors in tribal populations of Chhattisgarh state of India. Study can provide valuable information on Hypertension and its risk factors for state of Chhattisgarh. However manuscript have lots of deficiencies which confuses any reader. To draw your attention following are the observations: -

(1) In abstract, high-altitude has been associated with HTN, but manuscript does not support any data on altitude of study population/participants

(2) Key words - "Hypertension" and "HTN" both have been written

(3) In "Introduction" part of manuscript few sentences have written even without citing references. e.g. As such ........developing countries. in line 65-67 on page 4

(4) NO citation for "Existing...........................................Chhattisgarh." in line No. 70 - 72 on page 4.

(5) NO citation for "Currently ........................................tribal people." in line No. 72-73 on page 4.

(6) Reference 11 is about 42 years old, which is not relevant in current time

(7) Statement of undiagnosed HTN is provided in Line No. 82 - 83 on page 5. I wonder, whether undiagnosed HTN was research question of this study.

(8) Heading "Study population" on page 6 is includes many information about the data and methodology, which should not be included i defining the study population.

(9) A systolic BP has been wrongly defined as 121 - 90 mm Hg on line 128 on page 7

(10) Independent variables: Age groups are wrong

(11) Results: Line 154 page 8 - Of these 72010,27760 is confusing

(12) Table-1: Numbers of the categories of Age, Residence etc do not sum up to the size of participants depicted in column heading

(13) Table-2: Lots of data missing from the table. Total ST from urban and rural areas were 41.23% and 42.84% respectively - Data of 15.93% STs are not known. Similarly as per categories of Marital status i.e. "never married 40.91%; Currently married 44.6%; widowed 38.1% sup up to 123.61%, confusing.

(14) Table-3: Weighted % has been provided for STs and General population - Reason of taking weighted % not given in text

(15) Table-4: Many aORs having 1 in their 95% CI limits have been shown with asterix symbol, typically used to show significance level, are Not significant

(16) Line No. 214 - 217 on page 14 - " Another study by Kusuma et a...............prevalence of HTN. [15] - There was not such study in reference list authored by Kusuma

(17) Wrong citation method in line No. 256 on page 16

(18) Fig 1 provided in manuscript does not add value with respect to its various parts shown in map. Method of showing map of state may not be comprehensible to readers from world beside India

Reviewer #3: 1. The title should not include short forms

2. The study lacks clarity on distribution of sample population as indicated in various tables. There is also mismatch with the open data source in NFHS site, so there is need to spell clear reason for the same in methodology section.

3. source of core data should be mentioned in methodology

4. In table 3 Data, for better comparison data of tribal population should be compared with non tribal population (segregating tribal population for total population)

4. study results should be compared from similar literature regarding prevalence of hypertension among tribal vs non tribal population of central India

6. PLOS authors have the option to publish the peer review history of their article (what does this mean? ). If published, this will include your full peer review and any attached files.

**Do you want your identity to be public for this peer review?** For information about this choice, including consent withdrawal, please see our Privacy Policy .

Reviewer #1: **Yes: ** Nirmalya Mukherjee

Reviewer #2: No

Reviewer #3: **Yes: ** Suyesh Shrivastava

---

## [Author Response · Author response to Decision Letter 1]

19 Aug 2024

Rebuttal letter

Reviewer-1

1. A paragraph explaining why the focus is specifically on Chhattisgarh state. This will help contextualize the study's geographic scope.

Ans: Thank you for your comment. We have added more text per your suggestion to highlight the rationale. The changes can be seen on page ___121-130__, lines 73- 98

2. Given that numerous studies have already utilized NFHS 5 data to examine hypertension and its risk factors, add a paragraph that outlines the unique contributions of this research.

Ans: We have better highlighted the rationale in the introduction.

The changes can be seen on page __141-153___, lines 79-82

3. In the data analysis section, it is reported that no chi-square, mean, or standard deviation (SD) values were found using the respective statistical tests for categorical and continuous data. This needs verification and correction if necessary.

Answer – Thank you for highlighting this discrepancy. We have revised our statements under the data analysis section. All corrections done as per suggestion.

5. While Table 2 addresses the prevalence of prehypertension and Table 3 deals with hypertension, Table 4 only discusses hypertension without presenting an adjusted odds ratio for prehypertension. Given the high prevalence of prehypertension noted in the results, it would be beneficial to include a multivariate analysis for prehypertension in both sexes separately.

ANS- Given the high prevalence of prehypertension in the results, we considered your suggestion to include a multivariate analysis for both sexes separately. However, after careful consideration and internal discussions, we realized that a multivariate analysis for prehypertension was diluting our primary focus on understanding the factors associated with hypertension, and including a detailed analysis for prehypertension would extend beyond the scope of our current research objectives. Nevertheless, we have ensured that the prevalence and basic statistics for prehypertension are clearly reported in Tables 2 and 3, and compared with the total population estimates per NFHS to better understand the burden level specifically in the tribal population, which is otherwise diluted when presented as a part of total population.

Reviewer -2

(1) In abstract, high-altitude has been associated with HTN, but manuscript does not support any data on altitude of study population/participants.

Ans: We have revised our abstract to align with the changes done in the manuscript as per the comments from other reviewers.

(2) Key words - "Hypertension" and "HTN" both have been written

Ans: We have reframed our keywords based on MeSH terms.

(3) In "Introduction" part of manuscript few sentences have written even without citing references. e.g. As such........developing countries. in line 65-67 on page 4

Ans: References no 5 & 6 given for the lines 65-67.

(4) NO citation for "Existing...........................................Chhattisgarh." in line No. 70 - 72 on page 4.

Ans: references added for the line no 70-21, "Existing...........................................Chhattisgarh.".

(5) NO citation for "Currently ........................................tribal people." in line No. 72-73 on page 4.

Ans: Added citation for the same.

(6) Reference 11 is about 42 years old, which is not relevant in current time

Ans: Another reference has been added for the same meaning of text. The cited reference is at serial number_____11 only as it has been replaced.___

(7) Statement of undiagnosed HTN is provided in Line No. 82 - 83 on page 5. I wonder, whether undiagnosed HTN was research question of this study.

Ans: Undiagnosed hypertension is a major bottleneck of hypertension control. We have now mentioned this in the introduction section of the manuscript to throw some light on the burden of problem among tribal population. However, it was not further discussed in the manuscript.

(8) The heading "Study population" on page 6 includes much information about the data and methodology, which should not be included in defining the study population.

Ans: We have revised our methods as per your suggestions, and make it more readable and inferential.

(9) A systolic BP has been wrongly defined as 121 - 90 mm Hg on line 128 on page 7.

ANS. - We acknowledge the error in defining systolic blood pressure on line 128 on page 7, where it was incorrectly stated as 121 - 90 mm Hg. The correct definitions are as follows:

Prehypertension: Systolic BP of 120-139 mm Hg or Diastolic BP of 80-89 mm Hg

Hypertension: Systolic BP of 140 mm Hg or higher, or Diastolic BP of 90 mm Hg or higher.

We have corrected these definitions in the revised manuscript.

(10) Independent variables: Age groups are wrong

ANS. - We have updated the data analysis section, corrected the age group classifications to ensure all information is accurately reported.

(11) Results: Line 154 page 8 - Of these 72010, 27760 is confusing

ANS. - We have updated our results to make them more readable as per the suggestions.

(12) Table-1: Numbers of the categories of Age, Residence etc do not sum up to the size of participants depicted in column heading

ANS. - We have updated the data analysis section table-1, and corrected the results.

(13) Table-2: Lots of data missing from the table. Total ST from urban and rural areas were 41.23% and 42.84% respectively - Data of 15.93% STs are not known. Similarly, as per categories of Marital status i.e. "never married 40.91%; Currently married 44.6%; widowed 38.1% sup up to 123.61%, confusing.

Ans- In Table-2, we are presenting only the prevalence of pre-hypertension. The percentages represent the weighted proportion of the population who are pre-hypertensive. We are not displaying the total population figures here.

(14) Table-3: Weighted % has been provided for STs and General population - Reason of taking weighted % not given in text.

Ans: data has been corrected in the manuscript table 3 as per reviewer suggestions.

(15) Table-4: Many aORs having 1 in their 95% CI limits have been shown with asterix symbol, typically used to show significance level, are Not significant.

Ans: Corrected as per suggestions.

Reviewer #3:

1. The title should not include short forms

Ans: The title has been modified to convey the details of the paper in a more subtle manner as per reviewer suggestion.

2. The study lacks clarity on distribution of sample population as indicated in various tables. There is also mismatch with the open data source in NFHS site, so there is need to spell clear reason for the same in methodology section.

Ans: Data correction for all tables has been done. And analysis done again for the tables.

3. source of core data should be mentioned in methodology.

Ans: Mentioned. Source of core data is from NFHS-5 survey which was used for the separately for the Chhattisgarh. We have now revised our methodology, and mentioned details of data source early in methods section.

4. In table 3 Data, for better comparison data of tribal population should be compared with non-tribal population (segregating tribal population for total population).

Ans: Corrected as suggested by the reviewer. Comparison done for the tribal and non-tribal population. We have now added more reference to support our stance.

4. study results should be compared from similar literature regarding prevalence of hypertension among tribal vs non-tribal population of central India. We acknowledge the error in defining systolic blood pressure on line 128 on page 7, where it was incorrectly stated as 121 - 90 mm Hg. The correct definitions are as follows:

Prehypertension: Systolic BP of 120-139 mm Hg or Diastolic BP of 80-89 mm Hg

Hypertension: Systolic BP of 140 mm Hg or higher, or Diastolic BP of 90 mm Hg or higher

We have corrected these definitions in the revised manuscript.

Ans: Corrected

Other comments:

Q1. Please include your tables as part of your main manuscript and remove the individual files. Please note that supplementary tables (should remain/ be uploaded) as separate "supporting information" files.

Ans: Tables included as main part of manuscript.

Q2: Issue regarding copyright of Chhattisgarh map, also reviewer requested that there is no much need of to include Chhattisgarh map.

Ans: As we do not have copyright picture of Chhattisgarh map hence, we have removed from the manuscript.

---

## [Decision Letter · Decision Letter 1]

14 Jan 2025

Prevalence of Hypertension and its risk factors among tribal population in Chhattisgarh (India): Insights from the fifth round of the National Family Health Survey (2019-21)

PONE-D-23-38296R1

Dear Dr. Goel,

We’re pleased to inform you that your manuscript has been judged scientifically suitable for publication and will be formally accepted for publication once it meets all outstanding technical requirements.

Kind regards,

Pijush Kanti Khan, Ph.D.

Academic Editor

PLOS ONE

Additional Editor Comments (optional):

Reviewers' comments:

Reviewer's Responses to Questions

**Comments to the Author**

1. If the authors have adequately addressed your comments raised in a previous round of review and you feel that this manuscript is now acceptable for publication, you may indicate that here to bypass the “Comments to the Author” section, enter your conflict of interest statement in the “Confidential to Editor” section, and submit your "Accept" recommendation.

Reviewer #1: All comments have been addressed

Reviewer #4: All comments have been addressed

2. Is the manuscript technically sound, and do the data support the conclusions?

Reviewer #1: Yes

Reviewer #4: Yes

3. Has the statistical analysis been performed appropriately and rigorously? 

Reviewer #1: Yes

Reviewer #4: Yes

4. Have the authors made all data underlying the findings in their manuscript fully available?

Reviewer #1: Yes

Reviewer #4: Yes

5. Is the manuscript presented in an intelligible fashion and written in standard English?

Reviewer #1: Yes

Reviewer #4: Yes

6. Review Comments to the Author

Reviewer #1: (No Response)

Reviewer #4: Authors have addressed all the comments highlighted by the previous reviewers. Article may be accepted for publication.

7. PLOS authors have the option to publish the peer review history of their article (what does this mean? ). If published, this will include your full peer review and any attached files.

**Do you want your identity to be public for this peer review?** For information about this choice, including consent withdrawal, please see our Privacy Policy .

Reviewer #1: No

Reviewer #4: **Yes: ** Dr Pankaj Prasad

---

## [Editor Report · Acceptance letter]

PONE-D-23-38296R1

PLOS ONE

Dear Dr. Goel,

I'm pleased to inform you that your manuscript has been deemed suitable for publication in PLOS ONE. Congratulations! Your manuscript is now being handed over to our production team.

Kind regards,

on behalf of

Dr. Pijush Kanti Khan

Academic Editor

PLOS ONE